# Collective Properties of Trentino: From Traditional Competences to Modern Solution Providers

Alisia Tognon [1,2,*], Nicola Martellozzo [3,4] and Alessandro Gretter [4,5]

1. Dipartimento di Architettura e Studi Urbani (DAStU)-Politecnico di Milano, 20133 Milano, Italy
2. Laboratorio di Storia delle Alpi (LabiSAlp), Accademia di Architettura, Università della Svizzera Italiana (USI), 6850 Mendrisio, Switzerland
3. Department of Foreign Languages, Literature and Modern Cultures-University of Turin, 10125 Torino, Italy
4. Centro Universitario (GREEN) Groupe de Recherche en Education à l'Environnement et à la Nature, l'Università della Valle d'Aosta—Université de la Vallée d'Aoste, 11100 Aosta, Italy
5. Research and Innovation Centre-Fondazione Edmund Mach, San Michele all'Adige, 38098 Trento, Italy
* Correspondence: alisia.tognon@polimi.it; Tel.: +39-392-5691118

**Abstract:** This paper focuses on the context of the Autonomous Province of Trento in Northern Italy, which has hosted common regime institutions that manage collective mountain properties daily since the 13th century. These institutions operate in the most significant part of the territory and adapt their routines to emerging challenges. From different scientific perspectives (economic, anthropological, and architectural), we analyzed how this method has been actualized as the most effective management of local resources, generating opportunities for commoners, new citizens, and external users. This includes the exposure of the communities to novel economic activities, adaptation of the internal normative and planning systems, and reflection on how to combine natural resources with local needs and global scenarios.

**Keywords:** mountain commons; anthropological landscape; the Alps; medieval rules; Fiemme Valley

## 1. Introduction

We can consider the territory as a space inhabited by a human and a non-human community, circumscribed by the historical modeling process of these two communities. According to this definition, the commons represent a particular type of "getting in shape" process for the territory, as determined by collective "interest". This concept has adopted different meanings over time and has even been divided into different contexts, such as the mountain regions, where joint management and property regimes are often present [1]. During the medieval period, the interest in the commons was translated into a question of convenience and access to resources for the community [2–4], while, with the imposition of proto-capitalist and market dynamics (commercial companies, interregional exchange networks, and production planning), collective interest has since come to coincide with the economic one in a modern sense. Considering this development, the commons are the historical expression of a specific process that has shaped—and still shapes—the territory, according to a double process.

On the one hand, the landscape, conceived in its physical form, reflects the form assumed by the non-human community and represents the outcome of the processes and changes effected by the human community. On the other hand, the community dimension, framed as a social and family structure, is an expression of interests, negotiations, or tensions and pertains to the anthropological aspect. This social dimension translates to a "putting into practice" of the commons through an assembly of norms, customs, institutions, and laws in the juridical and economic dimension. These methods persist through mechanisms for the transmission of norms that characterize the community life, giving shape to the social dimension.

We can classify two levels within the common property regime, conceived as "institutional arrangements for the cooperative (shared, joint, collective) use, management, and sometimes ownership of natural resources" [5] (p. 27), of which one aims to define the "modality of use" of the commons, that is, to "regulate" community practices, while the other is believed to legitimize and guarantee the "possession" of community goods, that is, the modalities of "conservation". These two levels constitute the same practices of the human community in a specific way, namely, "practices of use" and "inheritance". Furthermore, the aspect of community management emphasizes these two characteristics, leading to greater involvement of the entire community. For example, alternation in pasture management aims to specify the present user and ensure the pastures' transmission (in a good condition) over the years. Similarly, the transmission of technical knowledge between generations aims to preserve (or innovate) traditional knowledge and enable its application in daily practices.

In this article, we focus on the mountains, which can be considered as global commons [6] and are territories that have been shaped through the sensible use of collective property, creating a context of extraordinary value. This historical and cultural process of the "use" and "modeling" (and modeling as use) of the commons must be observed from different points of view and perspectives of multidisciplinary fields. In particular, this paper highlights one example in the Alpine region, the Autonomous Province of Trento[1], which is a result of the centuries-old management of common lands [3] (Figure 1).

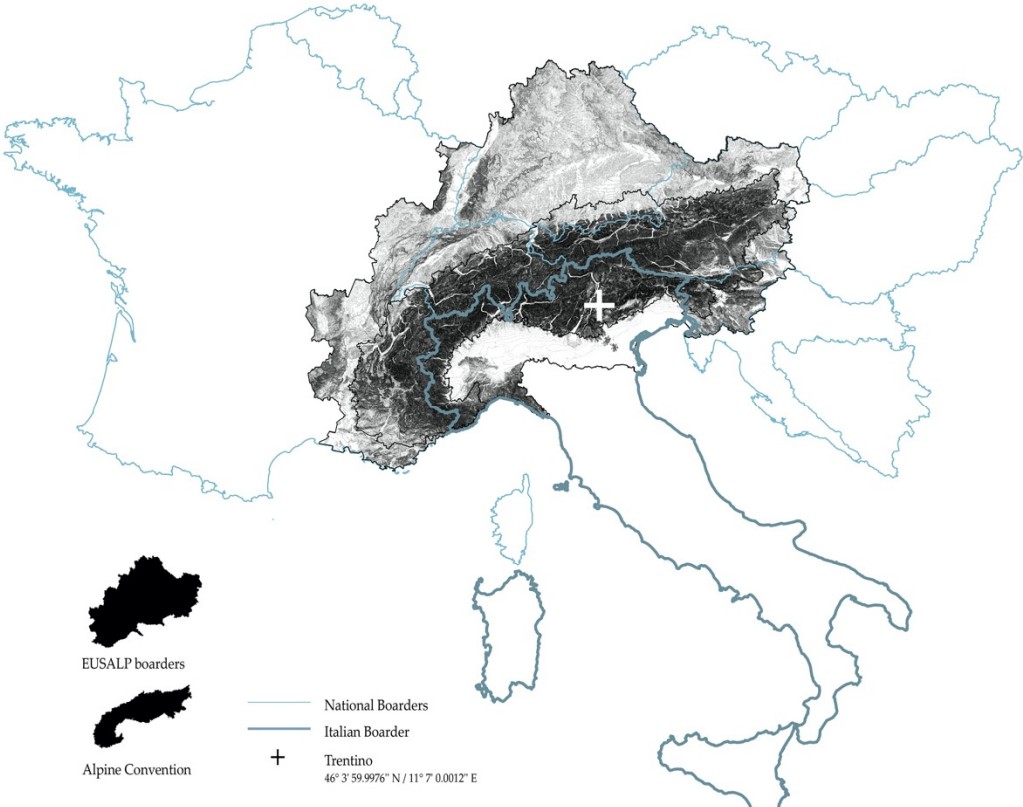

**Figure 1.** The map shows the location of Trentino within the Alpine region (in comparison with the EUSALP and Alpine Convention boundaries). Trentino is a predominantly mountainous province with a complex geomorphology. More than 70% of the entire territory lies 1000 m above sea level between the southern Rhaetian Alps and the Dolomites (highest peak of Cevedale: 3769 m). The morphology is characterized by a close network of valleys and a tangle of variously oriented and branching ridges, with flat strips of fluvioglacial excavation of the valley floors (source: "Europe" Geofabrick Download Server, graphic elaboration by the authors, 2022).

## 2. Material and Methods

The topic of collective properties among the research community and interested individuals in Italy has usually been considered by the historical or juridical sciences. However, the authors consider that this approach should be widened, including different sciences, new approaches and methods, and interrelations between individual fields.

As mentioned above, this research has been extended by introducing analyses in the field of economics (for budgetary analysis, business, and investment plans), statistics (for population, employment, and education analyses), and the social fabric and footprint permanence of the territorial organization of the landscape itself. Each research field must inevitably adopt different analytical approaches (data collection from primary and secondary sources, interviews, focus groups, shadowing activities...) and must include different actors (scholars, research centers, administrators, government representatives, and entrepreneurs, as well as a range of generations).

Moreover, the distinct research must include the past and present state of the collective property and the community and foresee future directions [7]. Following the specific disciplines, the authors systematized different models and methods of investigation.

Landscape explorations are considered fundamental for understanding, through spatial analysis, the dynamics which enable us to better understand and represent the regimes of cultural landscapes shaped by collective ownership. Historical plots are read by comparing how these palimpsests persist today. To achieve this goal, through the analysis of historical thresholds, graphically elaborated maps help us to highlight their permanence and absence, as well as the community's capacity to maintain and care for the territory and how collective properties have designed/altered/preserved the landscape in its spatial and morphological complexity.

Alongside mapping based on the archival map comparisons, as well as the data selected using the GIS of the Autonomous Province of Trento (PAT), the official reports of the PAT [8] were collected, catalogued, and summarized, and the interventions were carried out, providing relevant quantitative data on the entire region. Moreover, after almost two decades of applied research, the historical and juridical investigation has not ceased, analyzing original documents for the purpose of scrutinizing the most recently applied rules or decisions of the courts.

Even the research registered in the municipal databases, as well as the Magnifica Comunità di Fiemme's historical archives, allowed us to gather primary information in order to consider the centuries-old history of the Fiemme community. Considering the Magnifica Comunità di Fiemme as a case study, it is still one of the leading players in the valley, being the main forest owner and an essential stakeholder in any research in this specific territory. In the case of the Magnifica Comunità di Fiemme, the researchers organized several interviews with the leading social actors and the community to closely observe the ordinary and extraordinary forest management practices. In addition, despite the restrictions in Italy during the COVID-19 pandemic, the authors were able to carry out almost six months of ethnographic fieldwork (2020–21) in Val di Fiemme, focusing on forest restoration and reforestation.

### 2.1. Historical Literature Review

Based on our examination of the current studies and literature analysis, it is clear that the field of interest concerning this research has instead concentrated on the adaptation of the commons to recent economic changes [9–11]. In contrast, limited studies have analyzed their resilience, with respect to the ecological and territorial relationship, and society, with respect to its impact on rural areas [12,13]. Furthermore, from a spatial point of view the association remains unclear regarding their impact on the construction, care and conservation of the Alpine cultural landscape in a transtemporal manner.

In this identified territory, at the end of the 1960s, a renewed interest in these community governance systems emerged, with researchers analyzing some case studies in Italy [14,15] and Switzerland [16]. However, the 1990s research, with Ostrom's book as

a cornerstone [17], opened up new problems, on which basis a framework was drawn up to understand the perspectives of the common aspects of institutional and natural landscapes. Ostrom highlighted the positive differences concerning the most economically efficient public and private management solutions. These are based on social cohesion, creating well-being in communities through sustainable resource management practices and long-term visions fostering innovative solutions in response to the mountain territories' challenges and the management of the landscape as a whole.

Research has considerably moved beyond the concept of the "tragedy of the commons" [17,18]. Among the studies, De Moor [19–23] reviewed the historical basis of Hardin's theory [24] and combined extensive empirical analysis with explicit modeling and a highly developed theoretical framework involved in the development of innovative research methods.

However, a limited number of studies [25] have focused on the transformations and adaptations that have—or may have—undergone global economic and demographic changes at the local level, defining new tensions between the community and societal needs [26]. Although Bassi and Carestiato [10] showed that there are already attempts towards the bottom-up collective action of such organizational models, in agreement with Gibson-Graham et al. [13], they argued that the commons have become one of the tools for maintaining the ecosystem and consuming sustainable resources. Casari [27] and, more recently, Favero et al. [28] argued that Alpine communities, through the commons, have shown a real possibility of survival. In this direction, the studies still emphasized the importance of the regeneration of the regulations and rules that are used to interpret and efficiently manage the modern commons [29].

## 3. Context: The Study Area of Italy and Trentino

Common property regimes were predominant in Italy until the end of the 19th century and defined the land as inalienable, indivisible, unusable, and linked to agro-forestry-pastoral activities. In the 20th century, their dismantling began, to the extent that even today, the reference law at the national level is n. 1766 of 1926[2], which had as its objective the reorganization and liquidation of assets into individual properties. It was considered that this organization constituted an obstacle to the growth of the country and its medieval legacy and that, on the contrary, the freedom of ownership would guarantee a freer (and perhaps even excessive) exploitation of resources. On the contrary, the restrictions imposed by the organization of civic usages were an ancient attempt explicitly designed to prevent the depletion of environmental resources[3].

In Italy, collective properties (mainly forests and pastures) play a crucial role from an environmental point of view, as De Martin [30] and Merlo [31] highlighted. This issue makes it even more important to understand the historical management forms, given that the secular government of these properties has left a profound legacy both in the Italian land structure and the landscape, especially in the Alps and the Apennines. It is precisely in these mountainous territories, far from the large urban centers, that we find an intense concentration of collective properties under many names: *Consorterie* (Valle d'Aosta) [32], *Consortili* (Piemonte) [33], *Magnifiche Comunità* (Trentino[4], Friuli-Venezia Giulia, Veneto), *Consorzi vicinali* (Friulian Alps) [34], *Vicinie* (Lombardy) [35], *Comunalie* (Parma Apennines), *Regole* (Belluno and Vicenza Alps, Trentino) and the more recent Trentino ASUC[5] (Figure 2). Nevertheless, the plains and hilly territories, especially those in Central Italy, also host various collective properties linked to agricultural use, such as the *Università agrarie* (Lazio, Abruzzo), the *Partecipanze* (Emilia and Romagna plains) and the *Comunanze* (Marche, Umbria) [36,37].

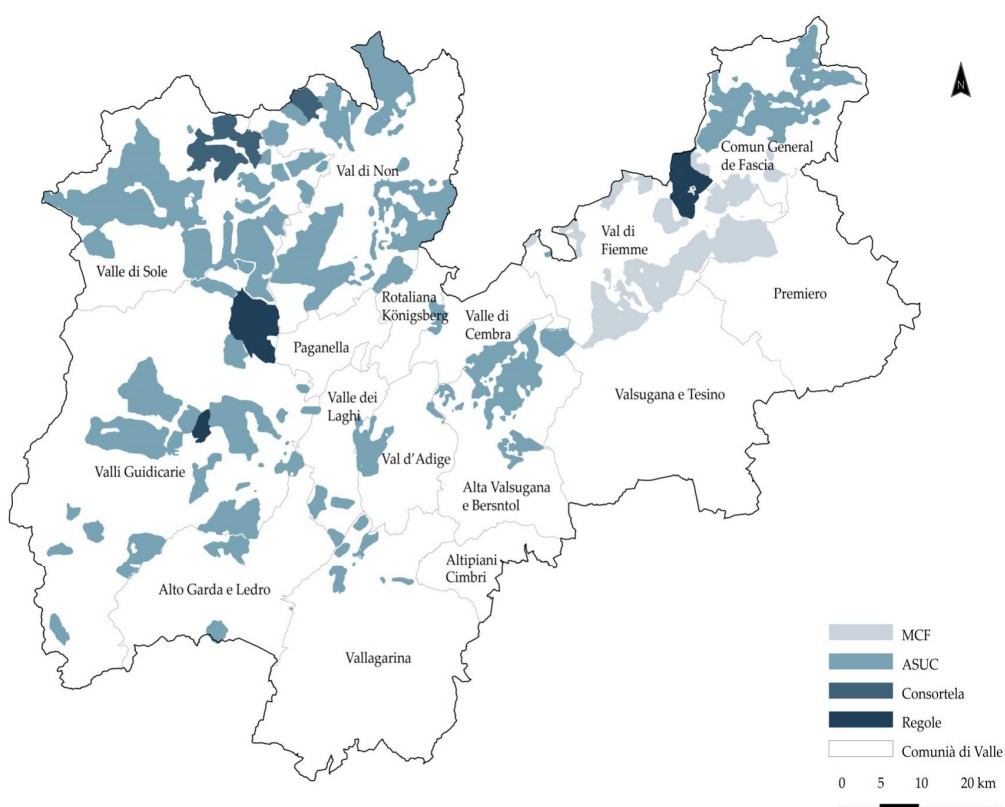

**Figure 2.** The map shows the partition of the Autonomous Province of Trento into the 15 Valley Communities, local territorial bodies that constitute the intermediate institutional level between the municipalities and the Autonomous Province. Established by Provincial Law no. 3 of 16 June 2006, the role of the Communities, as the associative bodies of the municipalities, is to represent the coordinating authority in making strategic decisions at the provincial level that reflect their impact on local realities. In addition, there are four types of Commons: the 'MCF' (*Magnifica Comunità della Val di Fiemme*); the 'ASUC' (*Amministrazioni Separate di Uso Civico* Trentine), born on 13 April 1987, amounting to approximately 102; the 'Consortele' in Val di Rabbi (today comprising 23 associations owning 5. 320 hectares of land and 'malghe'); the 'Regole' of Spinale and Mánez (in the municipalities of Ragoli, Montagne and Preore, an agrarian community existing *ab immemorial*), and the Regola Feudale of Predazzo.

An interesting fact concerns the agricultural area precisely classified as collective property, which in the last decade (2010–2020), according to the Italian Institute for Statistics (ISTAT), has decreased from 4.7% to 3.8% of the total area, that is, 482.000 hectares[6]. Moreover, the seventh ISTAT census found that 17% of this land is in Trentino Alto-Adige, representing almost half of Northern Italy [38,39]. However, while the agricultural area constitutes a decidedly relevant item of the collective properties of Central and Southern Italy, in Trentino, it represents a marginal part. The broader part of 1.5 million hectares classified as collective property (5.5% of the overall national surface[7]) is concentrated in the Italian Alps.

It is relevant to highlight that the above mentioned institutions are local expressions of collective property, and they obtained recognition in the Italian legal system with the law n. 168 of 20 November 2017 [40], with which the legal figure of the "collective domains" (*domini collettivi*) was established. This marked an important step, given that at the national level, the only existing framework was that of the 1927 law [41]. The Italian state recognizes collective dominion as "the primary legal system of the original communities", characterized by a collective heritage (natural, economic and cultural), a community of entitled persons and, above all, autonomy in the regulation of heritage management. In other words, the national framework concerns the landed level, while

the management aspect is delegated to the single collective domain, without a common reference.

Focusing on Trentino, it is evident that the tradition of commons is still a current reality in the organization of the mountains' legacy. The Autonomous Province of Trento and the Autonomous Province of Bolzano constitute the region of Trentino-Alto Adige/Südtirol in the northeast of Italy. In particular, the Trentino mountainous territory is composed of 166 municipalities, settled in an area of 6207 km$^2$ with a total population of 542,158 in 2021[8]. Geographically, Trentino is a corridor between the flat Padana valley, leading towards the Mediterranean Sea, and Central Europe, and culturally, it is still a hinge land between the Germanic and the Latin worlds today, which has also influenced the management of its natural resources, mixed between the roman experience and the German-rooted one [14]. This geographical position and its status as a crossroad between different cultures have influenced the cultural landscape and the socio-economic fabric of Trentino for centuries [42]. Economically, since the past era (14th–17th centuries), a relevant role has been played by the trade of wood exported to Venice and other territories for shipbuilding and construction, which has ensured the livelihood of the local populations, protecting against famine and deprivation [43]. Moreover, the region has a peculiar tradition in the silvopastoral sector, with a further focus on some agrarian cultures, such as grapes and apples, which constitute excellent products of Italy today (Figure 3). Due to the high peaks and the mountains' characteristics and beauty, Trentino has developed in the past century in the touristic and service sectors.

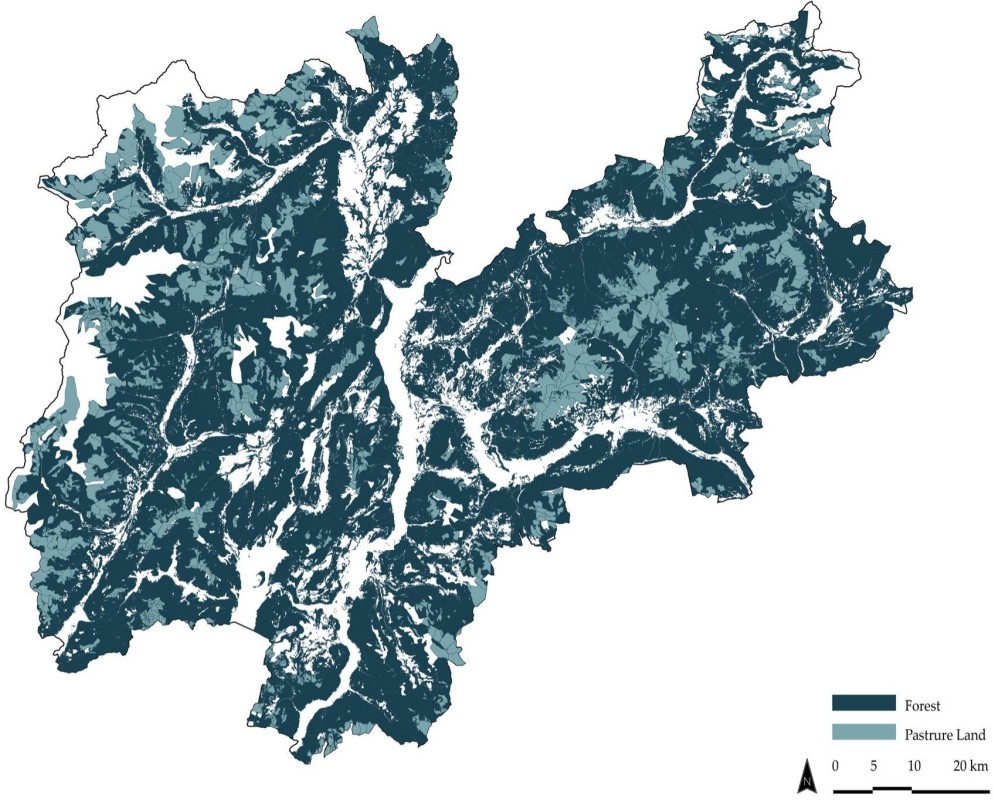

**Figure 3.** The maps show the spatial distribution, where most of the territory is covered by forests (390,463 hectares, 63% of the provincial surface: https://forestefauna.provincia.tn.it/Foreste, accessed on 25 September 2022), to which pasture meadows (about 35,000 hectares), which are usually collectively owned, have been added (Gis Database: Statistical Service of the Autonomous Province of Trento—data warehouse online, accessed on 25 September 2022. Graphic elaborations by the authors, 2022).

Land and resource ownership has also influenced the maintenance and development of the abovementioned peculiarities. In Trentino, more than 60% of the entire territory is

managed through forms of collective property. It represents the most significant example of this type of land governance in Italy, which is deeply rooted in the local culture and history [44]. Territorial and landscape planning and resource management are among the areas in which the self-government structure has most often been applied by innovating community-led practices for managing the commons, which date back to the medieval age [45,46] and were recognized by the Italian Constitution after World War II. Today's municipal geography of Trentino is still the result of the course of history and is dotted with arbitrations[9] among the municipalities and villages regarding the ownership of the mountains, which have established boundaries between the multiple "administrative islands"[10], forest districts, and pastures located outside the municipal limits.

## 4. Discussion from a Historical Perspective

### 4.1. Trentino's Historical Perspective

Considering that the experience of the collective properties has lasted for about ten centuries, the historical analysis can be divided into three different periods. The initial one is deeply connected with the Middle Ages, the second started just after the French Revolution, and the last is related to the post-war period.

As Salsa [47] stated, it is necessary to underline the highlands' peculiar ability to experiment with and develop political-administrative practices capable of enabling habitability at high altitudes. As reported by the *Carte di Regola* (Charters of Rule), the government of collective goods in the mountain territories has its roots in both Latin law (*res nullius*) and Germanic law (*Gemeinschaft*) [48]. However, the formalization of these rules began in the late Middle Ages, between the 12th and 14th centuries, through the written codification of 'ruling law'. The interest in regulating these lands occurred because of the paradigm shift in the consideration of the Alpine chain, which was then sparsely inhabited, from a place of simple crossing to a place of production [47]. Because of the exponential growth of the population and the awareness of the expansive extension of the plots of uncultivated land, the landowners began to regard them with interest.

From a historical perspective, the birth of many—or perhaps all—of the Trentino commons is linked to the need to affirm, maintain, and legitimize a precise method of use of these goods, namely the so-called *usi civici* (civic uses), which, during the Middle Ages, came to constitute real rights of use. In this sense, the interest in the commons will be maintained for as long as their use continues, including practices ranging from mowing and mooring to cutting wood [49]. Ortigalli, regarding the environmental protection measures of medieval communities, affirmed that "these concerns respond to the secure (and, yes, refined even if substantially material) perception of the intrinsic value of the natural good. A value that is protected as such, for its essence and quality even before the immediate benefit of a group, a class" [4] (p. 152).

Until the time of the Napoleonic government, the territory was administered by communities of villages called *Regola* ("Rules"), organized by the *Carte di Regola* ("Charters of Rule")[11]. These documents were based on ancient customs handed down in different places and, therefore, diversified according to the needs and uses of the communities. Their approval was submitted to the superior authority of the prince-bishop of Trento or the count of Tyrol, the local nobility and the governing bodies of the citizens. In the 11th century, and more extensively in the 12th century, the transition from the rights of use to common goods took place, in which the legal ambiguity of this category was resolved, and the ownership of the goods was defined [2]. In fact, the *Carte di Regola* consists of a formalization of the rights of use of the community and a recognition of the ancient customs by the territorial lord. Recognition, therefore, was granted, but as a concession from above. Nequirito [50] suggested that these pacts between the communities and the prince-bishopric (or dynasts) represented a way of linking these communities to the Trentino political mechanism. The *Carte di Regola* granted broad autonomy in the management of common goods, on which the bishop and the dynasts imposed tithes and other fiscal taxes.

These Rules created a conjunction between all the inhabitants, called "*vicini*", and the territory they inhabited: a conjunction between the people and territory (understood as a natural reality), the assembly of the heads of families and individual local realities with respect to the broader dimension of the valley. The administration of the Rules had the aim of controlling the normalized exploitation of common land and protecting small private, so-called "*divisi*" possessions. From the analysis of the transformative impact on the territory, the methods of use of natural resources have not changed over the years. They were based on the community exploitation of natural resources, while the cultivable plots belonged to the families. However, the definable private areas of the cultivated fields could also be subject to limitations in favor of the community [49].

The first half of the 19th century was marked by the Austrian Empire's centralization of state authority, accelerated by the Napoleonic Wars [37]. The result was the modernization of the feudal structure inherited from the Holy Roman Empire. The secularization of the Bishopric-Principality of Trento was completed in 1815, with its absorption by the County of Tyrol [51]. The administrative transformation also affected the management of the territory, with the consequent revision of the forestry regulations. The first act was the Forest law of 1852 [52], which, continued by the reform of the forest keepers in 1856, standardized forestry throughout the territory of the Austrian Empire. A second forestry organization reform soon followed (1859), introducing new offices, inspectorates and districts that remained substantially unchanged until Kirchlechner's law of 1895 [53]. However, the major change concerned the disappearance of the medieval Rules.

In most cases, these Rules were finally abolished with the Bavarian laws of 1807 and "transformed" into municipalities, but some found a way to perpetuate themselves and their heritage. The few surviving *Regole*, however, lost part (predominantly the most marginal one) of their patrimony in 1851, when the law on the relief of feudal burdens came into force. From the 13th century, it became standard practice for many *Regole* and *Magnifiche Comunità* to give farms, pastures or woods in their territory to private entities or other *Regole*, granting them the useful domain but keeping the direct one. With the law of 1852, all these assets became the property of the fiefdoms.

In any case, this historical *caesura* reopened the ancient question of the enjoyment of the community heritage by the inhabitants. The Napoleonic reform equated the *vicini* with all the other citizens: "the law implicitly created a distinction between two categories of citizens: outsiders, who were members of the municipalities but were excluded from the *Comunità Generale*, and insiders, who were heirs of the original members of the Community" [54] (p. 602). These reflections by Bonan explicitly focused on the *Magnifica Comunità* of the Fiemme Valley (the oldest and most extensive of these institutions in Trentino) but are valid for most of the cases in the Province [50]. The administrative transformation translated on a social level to an intra-community rift provoked by the new rights of citizenship, which essentially nullified any difference between the *vicino* (who enjoyed the benefits of the commons and owned them by right and descent) and the residents/foreigners, hitherto excluded from the law of proximity. This tension, in some cases, has perpetuated to this day, as in the case of the *Magnifica Comunità* of the Fiemme Valley, while in other cases, it has been overcome through a strengthening of identity, rigidly circumscribing the social group that owned the commons. Two examples, again in the Fiemme Valley, are the Feudo Rucadin and the *Regola Feudale* of Predazzo, which leveraged the notion of family belonging (through paternal descent) to maintain firm control of their commons despite the new historical situation. On the contrary, in terms of size and order, the *Magnifica Comunità* could not use such a clear-cut principle, and the question of belonging continued to recur until after World War II before interest in it was lost.

The new Italian Constitution granted specific autonomy to some regions in regulating the land laws. Among these, the Autonomous Province of Trento added a specific norm[12] that preserves the ancient rights connected to the collective properties. With the later regulations and dedicated regional law, the role of the oldest ones was institutionalized (for example, *Magnifica Comunità* of Fiemme in 1936 or Regole Spinale Manéz in

1960), and a specific frame was established in 1952 towards the creation of specific bodies (namely the ASUC—*Amministrazioni Separate degli Usi Civici*), or general indications for their maintenance were developed (mainly embedding the residual authority within the local municipalities) (Figure 2).

*4.2. Current Situation*

Along with the significant changes that occurred in the post-war period, it must be noted that from the 1960s (at least), a gradual disaffection with the body, or rather, the institution that manages the commons on behalf of the *vicini*, emerged. This disaffection can be read as the effect of an equally progressive fraying of the relationship between the *vicini* and commons to the extent that, today, local individuals are no longer interested in either its maintenance or its use, offering them a reinforcing negative loop. However, focusing on the aspect of the practices for "activating the commons", in our opinion, has several advantages. It highlights the crucial importance of the commons in the life of the Trentino communities while, at the same time, removing specific reinterpretations that unduly project ecological, sustainable or necessarily virtuous sensitivities onto the past. The commons exist to be used, albeit in a communal way, and this use allows them to persist over time. This use can sometimes lead to intense exploitation or impoverishment, but the collective dimension tends to be a form of balanced stewardship.

As mentioned above, the creation of the landscape is the result of a continuous relationship between the community and "environment". The commons' landscape has been modeled according to needs or purposes of the community, but recently, the drivers have become different. The commoners' direct needs have sometimes been irrelevant, and new investments (buildings, roads, recreational infrastructures, etc.) have been carried out more for external interests. Nevertheless, with direct revenue provided to the collective body, this has indirectly generated new opportunities and benefits for commoners and locals.

Finally, this historical perspective also lends itself to an opening up of the present and the future. The birth of the ASUC occurred in the first decades after World War II, together with the creation of new commons or the re-activation (or enhancement) of dormant realities, the creation of a specific federation and study centers, leading to a revival of interest which, certainly, was of a different order compared to the interests of the late medieval communities, but which, in any case, gave shape to these assets through specific practices. These practices have attained absolute actuality and are of the utmost relevance in shaping the future strategies of resilience for the mountain communities of Trentino.

After all, the commons exist to be used, albeit communally, and this use allows them to persist over time. Sometimes, this use can result in the intense exploitation of natural resources, or even their depletion, but the collective dimension allows the community to stop this from happening. By keeping this in mind, we can avoid falling into a representation that retrospectively extends ecological or virtuous sensitivities to the past. It is essential not to mythologize the commons in this respect: every form of ownership regime, whether public, private or shared, offers the owners the possibility to create valid institutions capable of managing the commons successfully in a sustainable way [55]. If anything, problems arise when the same resources are managed jointly by several institutions or when two or more regimes overlap. However, even a hybrid situation, which we often find in the Trentino area, does not necessarily translate into failure. Netting has already noted the balance between private and collective ownership in Törbel. For the American anthropologist, the tragedy of the commons was averted in this small Swiss village through democratic decision-making processes that prevented the excessive depletion of the common resources [16] (pp. 137–139), an essential lesson for today's institutions facing the difficult challenge of actualizing the commons.

## 5. Anthropogeographic Evaluation

In this section, we introduce a general analysis of the mountain landscape from an anthropogeographic perspective concerning the collective properties of Trentino. In the second part, spatial focus is placed on the oldest governing bodies of the collective properties in Trentino, concentrating on the *Magnifica Comunità* of the Fiemme Valley, settled in 1111, which is introduced using an anthropological method.

### 5.1. The Mountain as an Anthropogeographic Landscape

As Ingold defined it, the landscape is not simply a space or land and does not coincide with an excellent idea of 'nature'. On the contrary, the landscape "is the world as it is known by those who inhabit it, who inhabit its places and walk the paths that connect them" [56] (p. 156). Picking up the themes introduced at the beginning of this paper, the landscape is defined by the continuous 'acts of dwelling', which occur on the synchronic and diachronic planes. It is possible to intercept these constitutive relationships, which become an instantaneous materialization of a continuous process of overlapping and re-shaping. In this way, it is possible to "read" the traces of previous signs, "palimpsests" [57] or "taskscapes" [56]. "Taskscape" is the term used by Ingold to describe an interrelated and changing set of 'tasks', or constitutive 'acts of dwelling', where the landscape is a temporal materialization: "the temporality of the taskscape, while it is intrinsic rather than externally imposed (metronomic), lies not in any particular rhythm, but in the network of interrelationships between the multiple rhythms of which the taskscape is itself constituted" [56] (p. 160). Moreover, regarding these relationships between the human community and other living elements: "In its present form, the tree embodies the entire history of its development from the moment it first took root. Additionally, that history consists of the unfolding of its relations with manifold components of its environment, including the people who have nurtured it, tilled the soil around it, pruned its branches, picked its fruit, and—as at present—used it as something to lean against. The people, in other words, are as much bound up in the life of the tree as is the tree in the lives of the people" [56] (p. 168).

Therefore, communities, together with the bearer of their culture, ethnic roots and history (religious, political, social and economic), have defined a way of organizing life through customs and norms. The landscape should be viewed as a page on which the outcomes of history are imprinted. In mountainous areas in the Middle Ages, economic differences developed between the ancient Roman settlements and the newly settled Germanic agricultural areas. By comparing the two Germanic and Latin fields and considering the location and land characteristics, it is possible to identify how economic differences [58] and productive uses impacted the artificial construction of the landscape. (Figure 4) In the Roman-established territories, the ratio of arable land to livestock (the *Acker-Alp* system) was equivalent, and the south-facing slopes were worked up to the altitude limit of cereal cultivation on the terraces, where the crops were alternated [59]. The height-stratified cultivation system was blended with seasonal migration [60].

Due to the Roman hereditary system, the peasant population was concentrated in the settlements and the fractioned farms, which led to an increase in the number of inhabitants and a reduction in the area of the forests in order to extend the new productive land. However, the higher pastures were usually worked by the community.

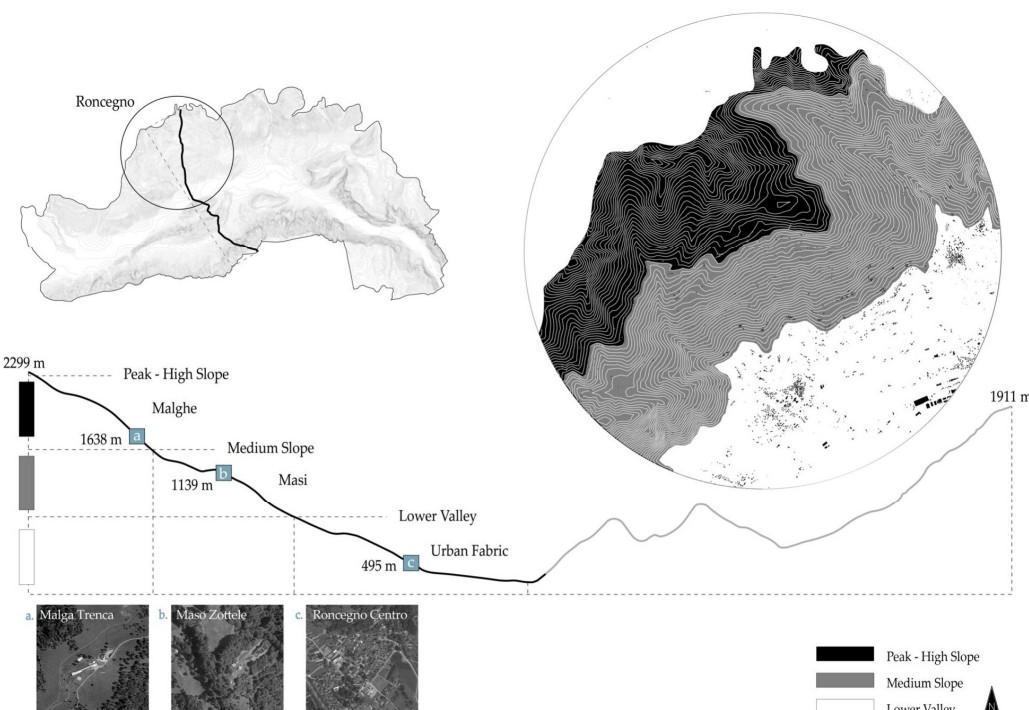

**Figure 4.** The section shows the three levels that make up the alpine system in the alpine mountains according to the organization of the land in the Latin tradition. According to the historical Latin model, the vertical infrastructure of the landscape is structured based on three nomadically inhabited levels. (1) The first level, including the valley floor, was the site of permanent settlements, crops and meadows. The farming family lived here in the winter with cattle in the stable. (2) The second level, the "maggengo", was characterized by temporary settlements and meadows. The farming family stayed with the cattle in the spring and autumn for the time required for haymaking operations. (3) The third level, on the peaks, was the alpine pasture or "malga", where the cattle were driven in the summer season. Each level was equipped with facilities and infrastructure, allowing the community to live and manage the animals and the landscape. This representation is fundamental to understanding the creation of the cultural landscape and the productive, ecological and social significance of alpine pasture systems. The section and the maps represent the territory of Roncegno in the Valsugana Valley, Trentino (© authors 2022).

Instead, in the northern regions, livestock farming was the most common subsistence economic activity, while cultivation was insufficient and took the form of field and pasture rotation. Silviculture was a crucial economic complement; even high pastures usually belonged to a single farm. This form of land use has been called the meadow-grazing system (*Wiesen-Alp* system) [61]. In the newly settled areas, especially in the Eastern Alps, farms remained subordinate to the clerical or lay owners. In these sites, the social structure was defined by the individual farm and the family, as a "unit", which played a significant role, while the agricultural society played a minor role. This model resulted in scattered settlements and isolated farms [62] (p. 87).

Agreement with the physical environment became an essential condition of the technical and social organization of the community. Even the relationships between geographic areas and communities, although affected by considerable spatial lengths, never result in processes of complete solitariness, but on the contrary, produce relationships of changing intensities, even between different altitudes of opposite-facing alpine slopes.

In this sense, the mountain landscape is a heterogeneous element physically made up of a diversity of shared and private goods, in which the change of scenery has been considered a tangible result of policies and market forces. As Cole and Wolf [14] (p. 65) studied, the political and economic relationship between different spheres and scales influenced the

local scale of the village, where the 'peasant' struggled to live and domesticate a problematic land. Indeed, in the more recently settled areas (*Jungsiedelland*), predominantly in the northern and eastern parts of the Alps, landlords granted the peasants tax privileges and greater rights, which extended as far as the election of their mayors and the management of lower levels of jurisdiction, due to the difficult conditions caused by the reclamation and cultivation of areas that were not yet colonized [17]. Nevertheless, these organizational approaches of collective management proved to be efficient in preserving natural assets, being based on a community capable of self-organizing and self-defining its rules for the use of local resources.

From a spatial point of view, it is interesting to note the ways in which these organizational patterns have become physically transliterated into spatial realms and, thus, into environmental and social palimpsests mnemonically rooted in local traditions and the ways in which the preservation of such practices over time has fostered the care of cultural heritage.

However, the specific transformations of the landscape from the Middle Ages to the contemporary age have not been adequately investigated thus far. There is also the difficulty of finding comparative and, evidently, non-overlapping maps. Only through the analysis of archives and, therefore, the numerical data or information written in the deeds offer us a chance to deduce, from the toponymy of places, their extent and their changes. It is important to emphasize that the relationship between the community and community property, whether private or semi-private, was based on the communal exploitation of natural resources. Cultivable plots of land belonged to families, although even the areas that could be defined as privately cultivated fields could be restricted for the benefit of the community.

The community's territory was organized according to four distinct levels: (a) the village, which represented the "private" aspect of each household; (b) the cultivated lands[13], usually located near the inhabited centers, which were made up of arable land and hay meadows and delimited through borders marked by hedges and fences or dry stone walls, with the whole community having the right of passage during the seasons of non-productivity; (c) the deciduous woodland[14] located in the areas surrounding the settlements, which was used for firewood, and the conifer woodland located in the higher areas, used to provide commercial timber; and (d) the pastures exploited collectively at different altitudes, which were used according to the seasons and presided over by communal "malghe"[15], in which the "malgari"[16], nominated by the community, produced the milk from the flocks or cattle on the pastures, and the shepherds were hired on behalf of the Rule.

In addition to these main elements, the community undertook the planning of the water system, which involved the maintenance of canals and embankments, the control of the purity of the springs, the regulation of the water flow and the exploitation of the resources through mills, sawmills, etc. Similarly, the road system was under compulsory maintenance by the community or frontier farmers. Thus, the *Regola* defined specific control regulations and severe punishments for those who damaged the roads and public places. At the same time, necessities, such as slaughtering or bread-making, and specific aspects of religious life were also regulated and managed by the *Carte di Regola*.

In light of the spatial analyses mentioned above, the centrality of the commons is justified. Throughout history, the commons have defined the landscape forms, whose overlapping signs can be seen even today. The commons have consistently demonstrated a high level of transformative adaptability, and today, an understanding of the past models and methods could contribute to the future development and regeneration of mountain territories. It has been highlighted that the transformations have occurred through community efforts and organization [12], which have cooperated in symbiosis with the natural mountain space. It is, therefore, necessary to read the historical palimpsests in order to define a common sense of needs for the purpose of current "habitat care". Their consti-

tution/maintenance is a necessary resource, as is their transformation, understood as a transformative act that is adaptable to current needs.

*5.2. An Anthropological Perspective on the Case of the Fiemme Valley*

The *Magnifica Comunità di Fiemme* stands out in the Trentino landscape for several reasons. It is the oldest collective property among those existing properties, manages an agro-forestry-pastoral heritage area of over 20,000 hectares on behalf of about 19,000 *vicini*, owns more than 12,000 hectares of certified forests [63–65], was among the commons hardest hit by the 2018 Vaia storm[17] (Figure 5), and is recognized by the Italian state as "an institution *sui generis*, a relic of ancient legal systems, which does not find exact correspondence in any of the categories of public bodies provided for in our system" [66]. The *Magnifica Comunità di Fiemme* occupies 33% of the Cavalese Forest District, but this value is increased to 60% if we consider the Fiemme Valley alone. To provide a comparison, the collective bodies under private law, such as the Regola Feudale of Predazzo, the Feudo Rucadin and the Vicinia Malgola (2700 ha), large private estates and the provincial state-owned forests of Paneveggio and Cadino (2750 ha), together, amount to 9% of the district [67] (pp. 27–28).

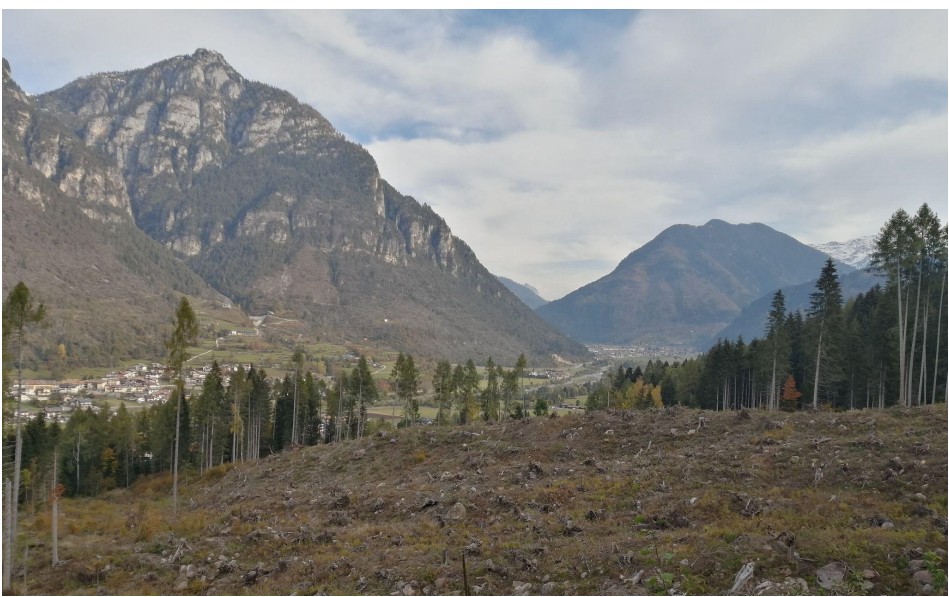

**Figure 5.** Large clearing near Ziano in the Fiemme Valley (Italy). In the background, a few larches that have survived the windthrow are visible (May 2020) (© authors 2022).

The *Magnifica Comunità di Fiemme* has changed the management modalities of its collective heritage several times, especially the forestry one. To summarize this change, the governance of the community forest patrimony has witnessed four distinct phases, comprising a first period (1270–1592) of occasional rules based on the usual methods, culminating in the issuing of the *Ordeni dei boschi* (Forest Laws). This second moment of the internal regulation of forest exploitation encountered a crisis during the second half of the 18th century, when the double pressure of the large commercial companies and the state power of the Austro-Hungarian Empire introduced a bureaucratic-scientific government of the woods [68]. The last turning point occurred with the abolition of the Rules, which occurred in 1807. Today's commons regime has inherited a situation made even more complex by the hybridization of public and private management forms [69], such as the contracting of work to external firms or the preservation of natural areas.

In the 1960s, a gradual disaffection began with the *Magnifica Comunità di Fiemme*, or the institution that manages the commons on behalf of the *vicini*. This disaffection can be interpreted as the effect of a progressive fraying of the relationship between the *vicini*

and commons to the extent that they are no longer interested in either its maintenance or its use. Part of the phenomenon is undoubtedly linked to the progressive devaluation of the forest heritage. As noted by Dalla Torre, "resources are no longer considered a source of sustainment for the community, so their collective management becomes unnecessary for local people, because they can rely on income derived from activities based on other resources and activities in other sectors. The resources therefore are left underused, abandoned, or become nostalgic symbols of the past identity of the territory, and community members do not feel engaged in the institutions anymore" [44] (p. 4). In the Fiemme context, tourism has replaced itself as an economic driver, especially in winter, which ensures that the valley has one of the highest occupancy rates in the region.

Unlike the other collective properties situated in the area, since the 1980s, the *Magnifica Comunità* has attempted to assign new value to its forest heritage. In 1997, it received the recognition of sustainability standards from the FSC®, becoming the first Italian forestry body, as well as the first body in the whole Alpine arc, to obtain this certification. In the last 25 years, the *Magnifica Comunità* has constantly reconfirmed the FSC® evaluations, and since 2007, it has also added the PEFC™ certification, thus obtaining a double recognition of its forests. Furthermore, two new initiatives were added to the existing certifications in response to the environmental and economic damage caused by the storm of 2018: the Chain of Custody of producers and the certification of eco-systemic services. We can consider the approach of the *Magnifica Comunità* as an attempt to re-economize the commons through a reorientation of the concepts of economy, community and nature. "The economy is conceptualized as the managing or negotiation (*nomos*) of habitat (*eco*). In this re-conceptualization of economy, commons function for the habitat maintenance to meet basic needs, support mutual well-being, consuming sustainably and distributing natural and social surplus to enrich social and environmental health" [44] (pp. 4–6).

The landscape, however, can be seen as a particular "shaping" of (and with) the non-human community [70]. To refer to the example of the Fiemme Valley, the combination of even-aged spruce forests, the scarcity of medium-altitude pastures and the widespread presence of the bark beetle contribute to the definition of the current landscape of the valley, with the woods that reach the borders of the human settlement stained by red "flames". However, each of these conditions can be read as the result of historical processes: the monospecific arboriculture of the 18th century, the gradual abandonment of pastures starting from the 1950s, and the proliferation of the bark beetle (*Ips typographus*)[18] in the aftermath of Vaia. We selected three phenomena characterized by different temporalities, including the long, medium and short term, to show how heterogeneous processes come into play in the formation of the landscape, being continuously juxtaposed with one another.

The forest certification strategy selected by the *Magnifica Comunità* represents an alternative way of "taking advantage of the void" compared to that considered by Viazzo and Zanini [71], which is not produced by demographic decline but linked to the crashes of Vaia. These voids, in both cases, constitute a crisis in the historical ways of living in the mountains. However, in the Fiemme Valley, this phenomenon has resulted in a specific re-evaluation of the centuries-old relationship between the human community and the forest. Storm Vaia heavily affected the landscape of the valley and, significantly, the woods most damaged were precisely those between the second and third levels; that is, that marginal area where trees have taken the place of pastures and cultivated land has been neglected or wholly abandoned. This is an example that demonstrates how the lack of practices for (and interest in) the management of the commons also reverberates and impacts on the continuous modeling of the landscape.

## 6. Insights and Final Considerations

Today, the current historical context highlights processes linked to various conflicts: the relationship between rural spaces and sprawl areas (residential, touristic, infrastructural, industrial), the private ownership and conflicts connected to the use of common utility resources (soil, water, forests...) and the acceleration of climate change, together with the

associated environmental risks. Furthermore, there is a loss of public significance concerning the public good, with the related decrease in the recognition of its value (economic, social and cultural).

With the urban hyper-exploitation [62] of the areas defined as "metro-mountainous" [72], the characteristics of cultural landscapes typical of many European countries have been modified, and territorial inequalities have been exacerbated, with the consequent creation of antithetical socio-demographic trends: the overpopulation of the lowest valleys together with the disrupted overlapping of the connectivity axes and, on the other hand, the depopulation of the medium slopes of the valley sections, as well as the more remote areas. The increasing pressure on the agricultural or (semi-)natural areas in the urbanized areas has also led to tensions and challenges, which must be framed in the context of the current development models, including infrastructural densification, migration, changes in lifestyles, shifting work activities and neglected traditions. All these factors have contributed to the anthropic territory's preservation (or loss). The management and maintenance of the cultural landscapes in Alpine areas have always required a strong human presence, guaranteed in past centuries by the network of collective properties, which operated as socio-ecological systems. The secular life of the *Regole* in Trentino has left visible traces on the current maps. The forms of land exploitation of the sylvan-pastoral assets are reflected today in the toponymic legacy of the sites, a memory of ancient customs and methods of Alpine resource utilization, and in the shaping of the landscape and the institutions still remaining today.

The ownership of the land, which belonged to a group of people with transmissible rights, turned it into private land, where single members enjoyed shared rights of exploitation. Moreover, the absence of a profit purpose ensured that the proceeds were reinvested for the benefit of the territory and community groups, providing economic support even in times of great crisis. Additionally, the idea of the co-ownership of a part of the municipality's land forced citizens to become aware of their responsibility concerning the care of the land they inhabited. The notion of environmental sustainability (understood as the ability to preserve natural resources over time without compromising them due to disorganized exploitation or lack of care), coupled with the social sustainability of controlling natural risks, would have led to the death of the entire society in such a hostile mountainous context.

Even 'virtuous' institutions such as the *Magnifica Comunità di Fiemme* have faced ongoing changes and new contemporary challenges imposed by climate change. Although the collective properties of the Fiemme Valley are an interesting case of rebalancing the relationship with the environment, the communal landscapes created over seven centuries of forestation need to be reconsidered today. The challenges caused by the Vaia storm with the falling of large stretches of high-quality forest and, later, the recent spread of the spruce bark beetle, have highlighted the unsustainability of specific historical modes of forest governance, including the communitarian one, which must, therefore, be redesigned. Knowledge inherited from the past no longer sufficiently guarantees the effective recalibration of the commons, but specific expertise remains a fundamental element for initiatives and projects in the territory. Over the past four years, the communities, external consultants and technicians of the *Magnifica Comunità* have debated the best strategy through which to 'modify' the landscape, e.g., by introducing new species and considering global warming scenarios.

On the one hand, Swiss experts proposed replacing spruces with Douglas firs, an allochthonous plant that is more resistant to heat and drought, based on the previous experience of the Vivian storm. On the other hand, the forest wardens of the *Magnifica Comunità* suggested natural regeneration based on decades of experience and in-depth knowledge of the territory. Finally, the Forestry Offices of the Autonomous Province of Trento appeared to prefer spreading a greater number of species to diversify the forest, even at the expense of its commercial value. We wish to emphasize that whatever strategy is selected, we are witnessing a reshaping of the Fiemme landscape, with the awareness

that this choice, initiated in today's world, will have a tangible effect for years to come. Moreover, as we can see from the example of the *Magnifica Comunità*, these associations are no longer the only body involved in decision making regarding their territory.

However, we must recognize that the knowledge inherited from the past is no longer sufficient to guarantee an effective recalibration of the commons. Nevertheless, specific skills remain a fundamental element enabling initiatives and projects to obtain a grip on the territory. We can mention, for example, the innovative role of some commons' governing bodies that aim to act as catalysts for development. In addition to attempting to manage their properties to the best possible extent, they look beyond the specific communities (e.g., the ASUC of Termenago). Today, many efforts aim to recompose overly fragmented private properties, which are usually located just outside villages, to implement initiatives for landscape conservation and defense against extreme hazards or, above all, to support new economic activities. In fact, small-scale initiatives such as animal husbandry and the cultivation of crops, herbs and fruits are working to reactivate abandoned and reforested terraces. The approach is highly relevant to the mountain territories of the Italian Alps, where the first barrier to the best governance and valorization of the land is unclear ownership. These bodies act as agents whose mission is to restructure the land patrimony, with the higher objective of keeping the traditional cultural landscapes active.

The question underlying this ongoing research is what the meaning of these institutions is today, as they are still a foundational aspect of individual empowerment, which affect the care of the land as a whole. Indeed, in several cases, history has witnessed how the loss of collective values resulted in the disappearance of entire communities and 'secularized' competencies of land uses. We must also ask how far the community should be involved in this process, considering what form of practical actualization of the commons could take place "by recognising new stakeholders participating in the negotiation process and transforming the use and place of the resource in the economy" [44] (p. 6).

These Trentino communities (as well as those in the rest of the mountain territories) should perhaps seek more pronounced forms of stewardship or collective custodianship, so that the neighbors and inhabitants can have a voice in internal decision making [73] (p. 37). This does not mean allowing people with no expertise or interests that are too partial to determine the operational modalities of land management (from forests to buildings) to be involved but, rather, ensuring that their representations, shared visions and desires find adequate space for expression. The objective of this ongoing research is to open up the discussion from a morphological-spatial and anthropological-social point of view on the commons, concretely verifying the status of an essential defining factor of the Trentino landscape and defining new tensions between the needs of the community and the demands of society [26] with respect to the landscape, understood as a unity and overall entity, for the definition of future guidelines.

**Author Contributions:** Conceptualization and investigation, A.T., N.M. and A.G.; Section 1, N.M. and A.T.; Section 2, A.T., N.M. and A.G.; Section 2.1, A.T. and N.M.; Section 3, A.T. and N.M.; Section 4, A.T., N.M. and A.G.; Section 5.1, A.T.; Section 5.2, N.M.; Section 6, A.T., N.M. and A.G.; original draft preparation, A.G. and A.T.; template editing, N.M.; resources, N.M. and A.T.; final review A.G. and A.T.; map visualizations and capture, A.T.; pictures N.M.; supervision A.T. and A.G. All authors have read and agreed to the published version of the manuscript.

**Funding:** This shared research was supported by the national research project "Territorial Fragility", DAStU (Department of Architecture and Urban Studies—Politecnico di Milano), awarded the title of "Excellence Department" by the Italian Ministry of University and Research (2018–2022); Laboratorio di Storia delle Alpi (LabiSAlp), Accademia di Architettura, Università della Svizzera Italiana (USI), Mendrisio, Switzerland; and the Doctoral School in Psychological, Anthropological and Educational Sciences (University of Turin).

**Institutional Review Board Statement:** Not applicable.

**Informed Consent Statement:** Not applicable.

**Data Availability Statement:** Not applicable.

**Acknowledgments:** A.G. is grateful for the support received by the administrators and commoners of Regole Spinale e Manéz for sharing their experiences and the presidents of the various ASUCs of Trentino, who attended meetings during the various in-field activities. A.T. is grateful to Brugger Robert (President of ASUC di Carbonare) for providing support with the research and for sharing data, as well as GianAntonio Battistel (Fondazione E. Mach).

**Conflicts of Interest:** The authors declare no conflict of interest.

## Notes

1.     The authors use the Autonomous Province of Trento or Trentino in the text to refer to the case study territory.

2.     L. 16 June 1927, n. 1766—"Conversione in l. del R. D. sul riordinamento degli usi civici". See: https://www.demaniocivico.it/leggi/nazionali/795-l-16-giugno-1927-n-1766, accessed on 4 December 2022.

3.     Many institutions started legal actions to protect themselves against this rule. See, for example, the "Cadore Rules Decree No. 1104/1948".

4.     In the text, the territory of the Autonomous Province of Trento, the official and legislative name, is replaced with the name of Trentino, which is the most used and recognized in Italy.

5.     https://www.asuctrentine.it, accessed on 27 September 2022.

6.     ISTAT 6th Agricultural Census (2010): https://www.istat.it/it/censimenti-permanenti/censimenti-precedenti/agricoltura/agricoltura-2010, accessed on 22 September 2022.

7.     According to the Agrarian Census of 2010, the amount is 1.668.851 hectares [35].

8.     https://www.provincia.tn.it, accessed on 4 October 2022.

9.     The parchments and documents kept in the municipal archives and the parishes are often dedicated to disputes between administrations regarding the laying of borders and positioning of crosses, which, in some cases, continue even today, making management and business investments more difficult in these properties.

10.     The administrative islands of Western Trentino are Pellizzano, Tassullo, Ronzone, Coredo, Malosco, Carisolo, Giustino, Strembo, Spiazzo, Massimeno, Ragoli, Stenico, Bleggio Inferiore, Breguzzo, Tione, Zuclo, Lardaro, Cimego, Riva, and Zambana. Those of Eastern Trentino are Baselga di Pinè, Pergine Valsugana, Calliano, Pieve Tesino, Cinte Tesino, Canal San Bovo, Imer, Soraga, Spera, and Lona-Lases [41] (p. 237).

11.     The "Charters of Rule" or *Carte Ordinamentorum* were documents that defined the exploitation of collective goods by the community, protected private property, appointed the competent administrative institutions and determined the ways of participating in the community life of the "vicini". Those who belonged to the community were defined as "vicini", while simple residents were considered "forestieri". Furthermore, the "vicini" were subjected to feudal rule but had autonomy in the governance of their economic activities. Initially, the statutes grouped ordinances handed down orally starting from the 13th century, and later they began to be transcribed, leading them to remain in use until the end of the 19th Century. They aimed to resolve disputes arising from the scarcity of resources and demanding environmental conditions in the mountain areas in the creation of rural societies and could concern individual villages and communities or include an entire valley [45] (pp. 15–23).

12.     The constitutional law of 26 February 1948, no. 5, which ratified the regional autonomy of Trentino-Alto Adige, empowered the region with legislative authority over civic uses. Separate administrations in the Province of Trento were subsequently regulated by D.P.G.P no. 4 of 11 November 1952, entitled 'Regulations for the execution of Provincial Law no. 1 of 16 September 1952 on the separate administrations of fractional civic use properties'.

13.     Moreover, part of the land to be cultivated was the "*fratte*": common lands provided to offer income to those who owned little land or those who had reasons of need, which, therefore, were rotated among the different "*fuochi*" (= family). They were also the lands left uncultivated and allocated to certain "fuochi" for their reclamation.

14.     Different local requirements regulated the exploitation of forests, and in some places, there was the custom of entrusting a small portion of land to a single family unit (*sorti*) [45] (p. 70).

15.     The word "*malga*" refers to the building, including the stable and the "*malgaro*" dwelling but also the surrounding livestock and pastures [45] (p. 82).

16.     "*Malgari*" is the Italian word originating from "*malga*" and is the person in charge of the guardianship and management of livestock grazing on mountain pastures (https://www.treccani.it/vocabolario/malgaro/, accessed on 4 December 2022).

17.     Storm Vaia was an extreme weather event that affected northeastern Italy (the mountainous area of the Dolomites and the Venetian Prealps) from the 26th to the 30th of October 2018. The storm brought extreme winds and persistent rainfall over the region. The winds reached 'hurricane' speeds (grade 12), blowing between 100 and 200 km/h for several hours. The persistent rainfall in the previous days, in addition to the wind, caused millions of trees to fall, destroying tens of thousands of hectares of alpine coniferous forests and resulting in a natural disaster [8].

[18]　It is a parasite insect, which mainly affects spruce trees (*Picea abies*) but also attacks *Picea orientalis*, *Picea jezoensis* and species of the genera *Pinus* and *Abies*. When a tree is attacked by the insect, which digs tunnels in the bark, the tree needles turn yellowish and then reddish-brown before falling off within a few weeks. The presence of the bark beetle can cause entire forests to become infested and die.

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
