# Peer review of "Collective Properties of Trentino: From Traditional Competences to Modern Solution Providers"

_land, doi:10.3390/land12010218_

Round 1

Reviewer 1 Report

This paper addresses an important aspect of so-called commons using a case in mountainous region of Italy. I believe that the contents described in English will be valuable to international audiences in order to know Italian commons cases that mostly had been written in Italy.

However, currently this paper is not well structured, or rather now seems to be very long descriptive sentences and it must be very difficult for readers to follow your discussion. It is understandable that it is difficult task to make this kind of paper in balanced well structure between historical literature reviews and some maps supporting discussion. Nevertheless, if authors improve and well structure in scientific manner this paper, this manuscript will drastically improved with higher values in this academic field.

I would suggest the following points.

1: Abstract

Now this abstract is not informative, mostly mention about research backgrounds. It must be improved to put tangible results and discussion parts.

2: Introduction

Particularly first paragraph is now without any reference. Later paragraphs also can be simplified and added references (especially ones in Italian) to reinforce your research standpoint.

3: Overall structures

As I mentioned, authors must follow basic scientific manner to write academic paper, although this manuscript include both historical reviews and mapping sections.

For instance I would recommend...

1 Introduction

2.Material and Methods

2.1. Study area (Italy and Trentino), 2.2.Historical literature reviews, 2.3.Supportive GIS contemporary mapping including past Trentino evidences

3.Results

3.1. Historical perspective

3.2. Current map and statistics indicating past situations

4. Discussion (current form might be applicable)

5. Conclusion

Figure 1: cannot understandable. put legend, azimuth, scale and meaning of cross mark etc. Please consider international audiences without deep knowledge about Italian contexts.

Also, I cannot find citation of this figure in the texts. it is also true for some other figures. I would strongly recommend improving these figures and use them to structure your discussion flow based upon above mentioned section structures. Literature and mapping must be distinguished.

Figures 2 and 3: Characters in Legend is too small to see, almost impossible to understand. Also, I would strongly recommend using colors making readers more understandable to this paper. This journal is online open access, so you can maximize its advantage to use feasible color figures.

Figure 3: Now you have TWO Figure 3s. you can combine then into one using colors correctly. One impressive map including adequate understandable information is essential in this academic field.

Figure 4: I think this is very important schematic landscape section profile. however, much information is lacking. use colors and bigger characters for readers, and need legend and scale of altitude and distance.

Author Response

Thank you for the important remarks that helped us to address more clearly our research topics:

According to your observations we have reviewed:

1. re-structured the entire paper (re-organization of chapters), according to the academic papers’ categorization, adding/reviewing/modifying some chapters.

2. Re-organized maps /pictures for supporting the discussion, with larger descriptions. We added information in the maps (legend, azimuth, scale and meaning of cross mark etc), to get more understandable even for an international audience, also adding colors. Moreover, in Figure 4 we added lacking info, used colors and bigger characters, legend and scale of altitude and distance. We checked to cite all the figures in the text.

3. Added more references where lacking, to reinforce the research standpoint.

4. The sentences and paragraphs have been simplified.

We hope to fit as our best all the remarks.

Reviewer 2 Report

The document is interesting and, importantly, it documents the historical processes that community property has experienced in the study area. The main contribution is the systemic vision of the process from all the data collection carried out. The lessons learned about the processes linked to this territory point to multiple strategies to be implemented to continue protecting all the resources linked to these areas of community property. The only element to be included in the conclusions of the study is how the findings would be linked to part of the existing literature in relation to cooperative property (shared, joint, collective), which would give added value to the study.

Author Response

Answer REV 1

Thank you for the important remarks that helped us to address more clearly our research topics:

According to your observations we have reviewed:

1. re-structured the entire paper (re-organization of chapters), according to the academic papers’ categorization, adding/reviewing/modifying some chapters.

2. Re-organized maps /pictures for supporting the discussion, with larger descriptions. We added information in the maps (legend, azimuth, scale and meaning of cross mark etc), to get more understandable even for an international audience, also adding colors. Moreover, in Figure 4 we added lacking info, used colors and bigger characters, legend and scale of altitude and distance. We checked to cite all the figures in the text.

3. Added more references where lacking, to reinforce the research standpoint.

4. The sentences and paragraphs have been simplified.

5. Throughout the whole paper we have tried to find the connection of the results with some of the existing literature in relation to co-operative ownership in Trentino

We hope to fit as our best the remarks.

Reviewer 3 Report

It has been a pleasure to read this paper. It presents the case of common-property regimes in Trento, providing a good approach, analysis and results in several levels, including economy, anthropology, and architecture. It is well reinforced by bibliographic resources, which have been well selected according to the content of the article. Also, the study of 3 separate historic stages helps understand the current situation. 

I would highly recommend to publish this article but there are some minor changes that must be done in order to compose a brilliant contribution for Land:

- Rewrite some sentences in order to link better one to another, especially in the introduction and conclusion. 

- There are missing references when referring historic facts. It could be stronger if the author provides some more references, especially in section 3. 

- Section 4 could easily be shorter. 

- Improve use of English. A second review by a copyeditor would definitely help to polish some structures inherited from the original language. 

In short, author provides a good context, and the article's goal is just that. Perhaps Section 4 and 5 could be sharper, but it is a good contribution as a first step of a further article on weaknesses and opportunities in that interesting Italian landscape. 

Author Response

Thank you for the important remarks that helped us to address more clearly our research topics:

According to your observations we have reviewed:

1. re-structured the entire paper (re-organization of chapters), according to the academic papers’ categorization, adding/reviewing/modifying some chapters.

2. Re-organized maps /pictures for supporting the discussion, with larger descriptions. We added information in the maps (legend, azimuth, scale and meaning of cross mark etc), to get more understandable even for an international audience, also adding colors. Moreover, in Figure 4 we added lacking info, used colors and bigger characters, legend and scale of altitude and distance. We checked to cite all the figures in the text.

3. Added more references where lacking, to reinforce the research standpoint.

4. The sentences and paragraphs have been simplified; some sentences have been rewritten in order to link better one to another in all the paper, as well we tried to be sharper in some passages.

5. Throughout the whole paper we have tried to find the connection of the results with some of the existing literature in relation to co-operative ownership in Trentino

6. we submitted our paper to a English proof reading, to polish some incorrect sentences.

We hope to fit as our best the remarks.

Round 2

Reviewer 1 Report

This is well improved and ready for publication. I would appreciate authors’ efforts.